# Unusual Domestic Source of Lead Poisoning

**DOI:** 10.3390/ijerph17124374

**Published:** 2020-06-18

**Authors:** Annamaria Nicolli, Grazia Genga Mina, Davide De Nuzzo, Isabella Bortoletti, Alberto Gambalunga, Andrea Martinelli, Fabiola Pasqualato, Mario Cacciavillani, Mariella Carrieri, Andrea Trevisan

**Affiliations:** 1Department of Cardiac Thoracic Vascular Sciences and Public Health, University of Padova, 35128 Padova, Italy; annamaria.nicolli@unipd.it (A.N.); ddenuzzo@gmail.com (D.D.N.); isabella.bortoletti@aopd.veneto.it (I.B.); alberto.gambalunga@unipd.it (A.G.); andrea.martinelli@unipd.it (A.M.); fabiola.pasqualato@unipd.it (F.P.); mariella.carrieri@unipd.it (M.C.); 2Department of Occupational Medicine, University of Rome Tor Vergata, 00133 Rome, Italy; 3EMG Unit, Data Medica Group, CEMES, 35128 Padova, Italy; mario.cacciavillani@gmail.com

**Keywords:** lead poisoning, lead migration, lead treatment, lead source, neurotoxicity

## Abstract

Non-occupational lead poisoning is not rare, mainly occurring in domestic situations in children, but also in adults. Lead poisoning was observed in a 65 years-old woman non-exposed to risk that caught our attention with a diagnostic suspicion of acute intermittent porphyria according to recurrent episodes of abdominal pain and neuropathy of upper limbs. Acute intermittent porphyria was excluded by a laboratory investigation that showed instead severe lead poisoning. After several thorough examinations of the domestic environment, the source of intoxication has been detected in some cooking pots that released high concentrations of lead. Ethylenediamine tetracetic acid disodium calcium therapy (three cycles) reduced consistently blood lead concentration and, after one year, neuropathy was almost entirely recovered.

## 1. Introduction

Lead is ubiquitous and occurs in the Earth’s crust in both organic and inorganic form. It is a soft, pliable, bluish-grey metal resistant to corrosion and belongs to the history of early metallurgy; it also owns antiradiation properties [1]. Because of all these features, it was one of the first metal used by civilizations dating back thousands of years. It occurs in the environment naturally and, to a greater extent, from anthropogenic activities such as mining and smelting, battery manufacturing, as petrol antiknock in the past, jewelry making, soldering, ceramics and leaded glass manufacture in informal and home-based industries. The prevalent way of exposure is by inhalation, but in the life environment by ingestion also. The prenatal exposure is relevant too, whereas absorption by skin is very small [2].

Lead inhibits three enzymes essential in haem biosynthesis: δ-aminolevulinic acid dehydratase (ALAD), coproporphyrin oxidase and ferrochetalase. For this reason, an important indication of chronic lead intoxication is sideroblastic anemia. Another important mechanism of lead toxicity is related to its characteristic of bivalent cation: it interferes with the role of other bivalent cations such as calcium, magnesium and zinc affecting renal, endocrine and neuronal functions and smooth muscles contraction. Clinical indications of lead poisoning are saturnine colic (acute), neuropathy, nephropathy, hypertension, encephalopathy and cognitive impairment, especially in children [3]. 

Many cases of lead poisoning are present in adults due to chronic occupational exposure, but due to the improvement of workplace hygienic conditions, it has become increasingly rare in industrialized nations [4]. Environmental and domestic intoxications in adults and in children also occur and are usually caused by the ingestion of lead-contaminated food or water.

In recent literature, several cases of domestic lead poisoning have been described, due to a large variety of causes such as lead pipes used for drinking water or the use of lead as a sweetener in home distillation of wine and rum [5], but also chronic use of glazed earthenware pottery [6] and retained bullets [7]. Moreover, in the last 20 years, cases of lead poisoning by traditional herbal medicine (Indian, Chinese, Ayurvedic remedies) [8] or cosmetics and by opium abuse appeared in the literature [9]. Hand-to-mouth activity remains a major route to lead exposure in children and the literature reports many cases of lead poisoning caused by pica-like behavior, by ingestion of plastic jewelry or by lead in housing paints, painted furniture or toys [10].

Absorbed lead binds erythrocytes (more than 99%) and finally the largest amount accumulates in bones. Liver is the soft tissue with the largest body burden of lead followed by kidneys, pancreas and other glands in less percentage. The liver is made of soft tissue and is most at risk to the adverse effects of lead, followed by the kidneys, pancreas and other glands of lesser percentage. Lead excretion is urinary and fecal and its half-life time is of 30 days in blood and 10 to 30 years in bones.

The case report describes a severe poisoning of a sixty-six year old woman due to a rare and unexpected source of lead.

## 2. Material and Methods

### 2.1. Lead Assay

Whole blood and urine (spot specimen collection) levels of Pb (PbB and PbU, respectively) were measured using graphite furnace atomic absorption spectrophotometer Perkin-Elmer AAnalyst 600 (Waltham, Massachusetts, USA) with Zeeman effect background correction (AAS). The detection limits (LOD) were a PbB of 0.7 µg/L and a PbU of 0.8 μg/L. The coefficient of variation was ± 5% and had an accuracy of 92–95%. The reagents and standard solutions used were high purity AAS grade. Standard stock solutions (1 g/L) of the metal were used to prepare the working standards throughout the analysis.

### 2.2. δ-Aminolevulinic Acid (ALA) and Porphobilinogen (PBG) Quantitative Assay

ALA and PBG quantifications by the Davis and Andelman method [11] were measured using a commercial kit (FAR Diagnostics, Verona, Italy). Urinary values were adjusted to creatinine determined applying the basic picrate Jaffe reaction.

### 2.3. Free Erythrocyte protoporhyrins (FEP) Quantitative Assay

The FEP blood concentration of was determined according to the fluorimetric method of Piomelli [12].

### 2.4. δ-Aminolevulinic Acid Dehydratase (ALAD) Activity Assay

ALAD activity in blood was determined by the Berlin and Schaller method [13].

### 2.5. Environmental Sampling

In order to evaluate the presence of lead in the air inside the patient’s flat, six area samplings of air were organized: the kitchen, the living room, the two bedrooms, the bathroom and the landing just outside the door of the flat. Samplings were carried out with an Institute of Occupational Medicine (IOM) sampler, certified UNI EN 13205: 2002 for inhalable airborne dust equipped with a cellulose ester filter (MCE) and connected to a pump with a constant airflow calibrated at 2 L/min. The samplings were performed placing the samplers on tripods at a height of 160 cm above floor level, and they lasted for an average time of three hours. The cellulose ester filters (MCE) have been analyzed according to the National Institute for Occupational Safety and Health (NIOSH) method n. 7082.

Samples of water from the four taps in bathroom and kitchen were also collected. The presence of lead on walls of the flat and on painting surfaces in the living room was evaluated by wipe tests following the NIOSH method n. 9105.

### 2.6. Migration Testing in Kitchenware

Migration testing on the 16 cooking pots were realized by putting glacial acetic acid 0.15 mL/cm^2^ into the pots for one hour under hood with laminar airflow. One hundred mL of milliQ water was put in cups and in plates and we heated them in the microwave with maximum power for 5 minutes. The Pb content in the water was quantified by AAS.

### 2.7. Treatment of Pills and Eyewash

The pills of melatonin and valerian were put in two beakers with 10 mL of milliQ water and left in rotating agitation for 30 minutes. After centrifuging at 1000 x g at RT, the supernatant was collected and analyzed by AAS. The eyewash drops were dosed as such.

## 3. Case Report

A 65 years-old Caucasian woman who did not consume alcohol, was a nonsmoker, and was working as an interpreter in a military base, was referred to our institute with the diagnostic suspicion of acute intermittent porphyria. The woman was suffering from recurrent episodes of abdominal pain that had started one year ago, deep asthenia, and neuropathy for six months (she could hardly walk by herself and she almost lost the use of her hands). Blood analysis of the past year showed only progressive sideroblastic anemia. No investigations had been made into a possible metal poisoning. According to diagnostic suspicion, the urinary excretion of porphyrins precursors such as ALA and PBG was measured. Due the values of ALA (45.90 mg/g creatinine; reference range (r.r.) < 4.5 mg/g creatinine) being significantly higher than those of PBG (4.42 mg/g creatinine; r.r. < 1.7 mg/g creatinine) and the suspicion that the symptoms could have been caused by lead poisoning, the metal was dosed in blood and in urine. The laboratory analysis confirmed the suspicion: lead in blood was 885 µg/L and in urine 285 µg/L.

Given the clinical picture and the high level of lead in blood, the patient was immediately hospitalized and chelation therapy with ethylenediamine tetracetic acid disodium calcium (EDTA) was started (1 g/die intravenous (i.v.) in saline for 5 consecutive days). Chelation revealed a high lead body burden with increased urinary excretion (1050 µg/L). Seven days after the last EDTA infusion, abdominal colic cleared up, but blood lead was 938 µg/L, so she had another cycle of EDTA (1 g/die i.v. in saline for 5 consecutive days). A month after the second chelation, the patient’s hemoglobin was within the reference range, but blood lead was still high (810 µg/L). She had a third cycle of EDTA (1g/die i.v., for five weeks, one dose each week): chelation was well tolerated with a normal hepato-renal function, and complete remission of anemia was observed. Table 1 summarizes the patient’s blood chemistry data.

Electromyography and electroneurography at the end of the third chelation showed signs of total bilateral denervation of the long radial extensor muscle of the wrist, of the common extensor muscle of the fingers and of the long abductor muscle of the thumb. The high-resolution ultrasonography of median, ulnar and radial nerves showed an increased cross-sectional area from 10 to 13 mm² (normal up to 6 mm²) of the radial nerve bilaterally at the spiral groove with a slight increase in dimension of nerve bundles, but without alteration on intraneural vascularization. The sensory branch of the radial nerve, ulnar and median nerve were normal. In conclusion, it was a bilateral severe axonotmesic injury of the motor branch of the radial nerve, localized right after the branches of the brachioradialis muscles. Clinical signs of injury were purely motor-related, characterized by a bilateral deep weakness of shoulders, arms and hands with early and severe involvement of wrist and finger extension.

Three months after the end of chelation cycles, a new electromyography showed evidence of bilateral subacute suffering of the long radial extensor muscle of the wrist (mild), of the common extensor muscle of the fingers (severe), with a complete bilateral denervation of the long abductor muscle of the thumb. The ultrasonography showed a bilateral decrease in the cross-sectional area of radial nerve (7–9 mm²). There still was a bilateral severe axonotmesic injury of the motor branch of the radial nerve, localized right after the branches of the brachioradialis muscles. At this time, bilateral recovery was very good but still partial: the patient quickly regained the strength of her shoulders and arms, hands and fingers, which she could move normally, except for the thumb.

Eight months after the end of chelation, electromyography showed evidence of bilateral subacute suffering of the long radial extensor muscle of wrist (mild), of the common extensor muscle of fingers (moderate) and of the long abductor muscle of the thumb (severe). The ultrasonography showed a bilateral decrease in the cross-sectional area of radial nerve (6–8 mm²). A bilateral axonotmesic injury of the motor branch of the radial nerve was confirmed, localized right after the branches of the brachioradialis muscles. The recovery of the hands was almost complete with improved movements and strength of thumbs about one year later (Figure 1).

The patient’s history did not reveal any professional lead exposure and there was no evidence of it, and nobody else at her workplace ever had signs or symptoms of saturnism. Therefore, an investigation of her everyday lifestyle and then an inspection in her flat for an environmental sampling of the air was planned in order to look for any object that could suggest lead contamination (dishes, cups, pots, drugs, water, etc.).

An accurate evaluation of pottery and pots (pots were the unique source of lead), is showed in Table 2. On the contrary, lead was not found in herbal medicines (valerian and melatonin), tap water from the kitchen and the bathroom, or eyewash (Table 3). Furthermore, wipe tests performed on the walls and paintings did not detect lead (data not shown). Finally, the evaluation of the indoor environment (Table 4) excluded lead pollution.

Migration testing were realized as above.

High migration as a result of pottery numbers 1 and 2 releasing high lead concentrations is likely the cause of lead poisoning, as the patient’s mother and caregiver also showed signs of abnormal lead absorption, albeit to a lesser degree due to the shorter period of using the pots.

The lead concentration in the blood and in urine of people who lived next to her flat was in the control range. The patient’s mother and caregiver showed lead increase in blood (360 µg/L and 240 µg/L, respectively) without any clinical signs or symptoms.

## 4. Conclusions

Severe lead poisoning is not rare in non-occupational exposure. These events are more frequent in children due to several causes such as tin pots, tinfoil and folk remedies [14], or lead-based paints in children PVC toys [15]. Atypical sources of lead poisoning in children were reviewed some years ago [16]. In adults, it is less frequent and is mainly linked to the use of herbal remedies [5,7,8]. Food (meat, probably as in the case described due to release of metal by pots) or vegetables from soils polluted by industrial discharges (not only Pb, but other heavy metals such as cadmium and mercury) can be a source of toxicity. Fortunately, they rarely reach toxicity levels such as those described. The population is generally unaware of the dangers inherent in certain articles of dubious origin; it is therefore necessary to buy certified goods (products of any kind—edible and inedible).

The underestimation of the symptoms, especially among non-experts in the field, leads to a delay in diagnosis. This case report is certainly relevant because it could be a resource for new occupational medicine residents who are beginning to learn about historical occupational diseases, but it is also the subject of lessons and seminars for medical school students. A specialized laboratory and clinical experience as a consequence of confidence with this kind of pathology allowed us to find, even if late in diagnosis, the therapy and a good resolution of the case.

The loss of confidence with lead due to a substantial reduction of this risk in developed countries is the cause of non-diagnosed poisoning, especially in subjects not-occupationally exposed to the metal.

## Figures and Tables

**Figure 1 ijerph-17-04374-f001:**
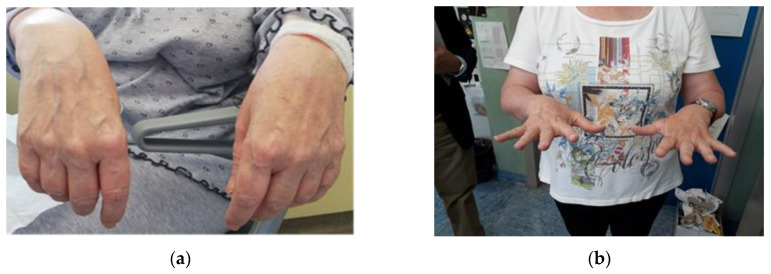
In July 2018, the patient’s hands were extremely weak and she could not extend her hand on the wrist (**a**). Nine months later, in April 2019, she obtained complete resolution of the neuropathy as a result of chelation therapy and rehabilitation (**b**).

**Table 1 ijerph-17-04374-t001:** Results of blood and toxicological test before and after every cycle of therapy.

	Admission	EDTA	Control *	EDTA	CONTROL †	EDTA	Control ‡
	07/24/2018	07/31/2018	08/07/2018	08/16/2018	09/26/2018	11/07/2018	02/06/2019
Hemoglobin (g/L)	88	91	105	118	108	n.d.	n.d.
Hematocrit (%)	30	28	34	28	36	36	40
Pb-B (µg/L)	885	540	938	400	815	525	425
Pb-U (µg/L)	285	185	521	196	317	178	59
ALA (mg/g creat)	45.9	26.1	91.4	5.7	54.2	n.d.	14.3
PBG (mg/g creat)	4.2	2.5	4.2	0.5	1.7	n.d.	1.4
ALAD (U/L)	15.7	11.9	1.8	2.9	n.d.	n.d.	3.5
FEP (µg/L)	1488	856	1256	506	619	2459	1352

Legend: On 08/20/2018 the patient was discharged from the hospital. * 7 days after the 1st EDTA infusion, **†** 30 days after the 2nd EDTA infusion, **‡** 30 days after the 3rd EDTA infusion. n.d. = not determined.

**Table 2 ijerph-17-04374-t002:** Migration test on cooking pots and pottery.

pot #	Inner Material	Lead µg/L	Lead µg/cm^2^
1	teflon	187.20	2.163
2	teflon	303.05	2.681
3	teflon	35.07	0.338
4	teflon	22.52	0.112
5	teflon	45.96	0.278
6	teflon	18.18	0.080
7	teflon	6.67	0.026
8	teflon	2.05	0.014
9	teflon	2.10	0.009
10	teflon	0.00	0.000
11	teflon	1.66	0.004
12	pottery	25.82	0.146
13	pottery	4.74	0.027
14	pottery	71.21	0.431
15	pottery	12.20	0.043
16	stainless steel	17.28	0.121

**Table 3 ijerph-17-04374-t003:** Lead content in herbal medicines (valerian and melatonin), tap water (kitchen and bathroom) and eyewash.

Sample	Lead µg/L
Kitchen water tap	<0.8
Bathroom water tap	<0.8
Shower water tap	<0.8
Valerian pills	<0.8
Melatonin pills	<0.8
Eyewash	<0.8

**Table 4 ijerph-17-04374-t004:** Lead measurement in the indoor environment (TLV-TWA 0.05 mg/m^3^).

Room	Lead mg/m^3^
Kitchen	<0.0000141
Living room	<0.0000141
Bedroom 1	<0.0000141
Bedroom 2	<0.0000141
Bathroom	<0.0000141
Landing	<0.0000141

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
