# Peer review of "Unusual Domestic Source of Lead Poisoning"

_ijerph, 2020, doi:10.3390/ijerph17124374_

Round 1
Reviewer 1 Report
It is better if the authors consider the following mentioned remarks and further improve the manuscript before submitting the final version.
- In the results section, the testing results of environmental samples, such as: air, waters and pills must be listed and analyzed.
- The Pb concentrations in blood of patient’s mother and caregiver (360 µg/L and 240 µg/L,200 respectively) were reached mild to moderate levels of intoxication, indicating that “food ingestion” might be the main expousure pathway of lead. So, the Pb contents in the staple food (rice, wheat and vegetables) must be taken into consideration in this study.
- It is better to put the sentence of “The patient issued a written consent to the anonymous publication of the case” in the section of acknowledgement.
- The “conclusion” section should be rewrite. I think the domestic source of lead poisoning and measures to prevent should be emphasized.
Author Response
Reviewer 1
Open Review
(x) I would not like to sign my review report
( ) I would like to sign my review report
English language and style
( ) Extensive editing of English language and style required
( ) Moderate English changes required
(x) English language and style are fine/minor spell check required
( ) I don't feel qualified to judge about the English language and style
|
Yes |
Can be improved |
Must be improved |
Not applicable |
|
|
Does the introduction provide sufficient background and include all relevant references? |
(x) |
( ) |
( ) |
( ) |
|
Is the research design appropriate? |
( ) |
( ) |
(x) |
( ) |
|
Are the methods adequately described? |
(x) |
( ) |
( ) |
( ) |
|
Are the results clearly presented? |
( ) |
( ) |
(x) |
( ) |
|
Are the conclusions supported by the results? |
( ) |
( ) |
( ) |
(x) |
Comments and Suggestions for Authors
It is better if the authors consider the following mentioned remarks and further improve the manuscript before submitting the final version.
Reply: we are grateful to the reviewer for suggestions to improve the manuscript.
- In the results section, the testing results of environmental samples, such as: air, waters and pills must be listed and analyzed.
Reply: according to suggestion of the reviewer, we implemented this section with two tables (3 and 4) on Pb content of air, water and so on.
- The Pb concentrations in blood of patient’s mother and caregiver (360 µg/L and 240 µg/L,200 respectively) were reached mild to moderate levels of intoxication, indicating that “food ingestion” might be the main exposure pathway of lead. So, the Pb contents in the staple food (rice, wheat and vegetables) must be taken into consideration in this study.
Reply: mother and caregiver lived in that period with the patient, so they ate cooked food in the "indicted" pots. The food was purchased in supermarkets or in groceries. It did not come from private farms or gardens, given that the family lived in a condominium in the downtown. There were also no sources of pollution related to the presence of a nearby industrial area.
- It is better to put the sentence of “The patient issued a written consent to the anonymous publication of the case” in the section of acknowledgement.
Reply: the sentence was put in the section of acknowledgement, as suggested.
- The “conclusion” section should be rewrite. I think the domestic source of lead poisoning and measures to prevent should be emphasized.
Reply: the conclusions were rewritten emphasizing the relevance of domestic poisoning, especially in adults and the importance of this case report as teaching for occupational medicine residents.

Reviewer 2 Report
The manuscript “ Domestic Unusual Source of Lead Poisoning ” (No.824460.)by Annamaria Nicolli described on laboratory experiments that investigate the released high concentrations of lead. The case report tells about a severe poisoning of a sixty-six year old woman due to a rare and unexpected source of lead. It is very important to lead poisoning in relevant environment for human health. The case report is very typical, However, also have some problems need to be improved.
1.There are similar between line28-32 and line 206-211, please improve it.
- I suggest that rewrite the conclusions.
Author Response
Reviewer 2
Open Review
(x) I would not like to sign my review report
( ) I would like to sign my review report
English language and style
( ) Extensive editing of English language and style required
( ) Moderate English changes required
(x) English language and style are fine/minor spell check required
( ) I don't feel qualified to judge about the English language and style
|
Yes |
Can be improved |
Must be improved |
Not applicable |
|
|
Does the introduction provide sufficient background and include all relevant references? |
(x) |
( ) |
( ) |
( ) |
|
Is the research design appropriate? |
(x) |
( ) |
( ) |
( ) |
|
Are the methods adequately described? |
(x) |
( ) |
( ) |
( ) |
|
Are the results clearly presented? |
( ) |
(x) |
( ) |
( ) |
|
Are the conclusions supported by the results? |
( ) |
(x) |
( ) |
( ) |
Comments and Suggestions for Authors
The manuscript “ Domestic Unusual Source of Lead Poisoning ” (No.824460.) by Annamaria Nicolli described on laboratory experiments that investigate the released high concentrations of lead. The case report tells about a severe poisoning of a sixty-six year old woman due to a rare and unexpected source of lead. It is very important to lead poisoning in relevant environment for human health. The case report is very typical, However, also have some problems need to be improved.
Reply: we are grateful to the reviewer for suggestions to improve the manuscript.
1.There are similar between line28-32 and line 206-211, please improve it.
Reply: The sentence in the abstract has been lifted and in the conclusions has been partially rewritten.
- I suggest that rewrite the conclusions.
Reply: the conclusions were rewritten emphasizing the relevance of domestic poisoning, especially in adults and the importance of this case report as teaching for occupational medicine residents and medicine students.

Reviewer 3 Report
First I want to congratulate the authors on a great case study. I recommend that you collaborate with a native English speaker to increase the impact of this amazing case study. At times it was difficult to follow where the authors were going and by fixing these issues it will make this case study one that Universities can use for examples of abnormal exposures to make their students think outside the box.
On line 48 you have his and I think you meant to have this
Author Response
Reviewer 3
Open Review
(x) I would not like to sign my review report
( ) I would like to sign my review report
English language and style
( ) Extensive editing of English language and style required
(x) Moderate English changes required
( ) English language and style are fine/minor spell check required
( ) I don't feel qualified to judge about the English language and style
|
Yes |
Can be improved |
Must be improved |
Not applicable |
|
|
Does the introduction provide sufficient background and include all relevant references? |
(x) |
( ) |
( ) |
( ) |
|
Is the research design appropriate? |
(x) |
( ) |
( ) |
( ) |
|
Are the methods adequately described? |
(x) |
( ) |
( ) |
( ) |
|
Are the results clearly presented? |
(x) |
( ) |
( ) |
( ) |
|
Are the conclusions supported by the results? |
( ) |
(x) |
( ) |
( ) |
Comments and Suggestions for Authors
First I want to congratulate the authors on a great case study. I recommend that you collaborate with a native English speaker to increase the impact of this amazing case study. At times it was difficult to follow where the authors were going and by fixing these issues it will make this case study one that Universities can use for examples of abnormal exposures to make their students think outside the box.
Reply: we hope that the English has been improved. In the conclusions, we emphasized the relevance of domestic poisoning, especially in adults and the relevance of this case report for students and residents to approach difficult clinical cases with due competence.
On line 48 you have his and I think you meant to have this
Reply: we apologize, the possessive adjective his has been improperly used instead of its. The text has been corrected.

Reviewer 4 Report
This is an interesting case report on household accidental lead poisoning. The paper is relevant for the journal, in my opinion. There are several improvements the authors should tend to.
Major comments
1) The methods used for lead quantification are described only for the patient's biomedia and not for e.g. water, kitchenware etc. This should be described in brief in the paper.
2) Quality assurance measures for Pb quantification has to be presented if applicable.
3) It would be interesting if the authors provide some information on the source of the lead-contaminated kitchenware in their paper, was it purchased on the standard market or had some stranger origin.
Minor comments
The paper requires a minor language correction. Also, please pay attention to the use of either British or American spelling standard, now there are some traits of both.
The sentence "This case-report is a paradigm and only the knowledge associated with a specialized laboratory and clinical experience, consequence of confidence with this kind of pathologies, allowed us to find, even if late in diagnosis, the therapy and a good resolution of the case" present in abstract and conclusion is magniloquent and unclear.
L21 non exposed should be together or with a hyphen.
L38 I guess lead is known from ancient times since it can be quite easily produced from the ore by the early history metallurgists and not because of the fact it is a bad conductor etc.
L45 a typo, haem
L121 herself doesn't need a hyphen.
L180 Avoid using contraction in an academic text (could not).
L186 lifestyle without a hyphen.
L190 Revise the sentence since now it is misleading and sounds as you have pots rather than data in Table 2.
Author Response
Reviewer 4
Open Review
(x) I would not like to sign my review report
( ) I would like to sign my review report
English language and style
( ) Extensive editing of English language and style required
(x) Moderate English changes required
( ) English language and style are fine/minor spell check required
( ) I don't feel qualified to judge about the English language and style
|
Yes |
Can be improved |
Must be improved |
Not applicable |
|
|
Does the introduction provide sufficient background and include all relevant references? |
(x) |
( ) |
( ) |
( ) |
|
Is the research design appropriate? |
(x) |
( ) |
( ) |
( ) |
|
Are the methods adequately described? |
( ) |
( ) |
(x) |
( ) |
|
Are the results clearly presented? |
( ) |
(x) |
( ) |
( ) |
|
Are the conclusions supported by the results? |
(x) |
( ) |
( ) |
( ) |
Comments and Suggestions for Authors
This is an interesting case report on household accidental lead poisoning. The paper is relevant for the journal, in my opinion. There are several improvements the authors should tend to.
Reply: we are grateful to the reviewer for suggestions to improve the manuscript.
Major comments
1) The methods used for lead quantification are described only for the patient's biomedia and not for e.g. water, kitchenware etc. This should be described in brief in the paper.
Reply: as suggested, we explained as Pb in other media was measured.
2) Quality assurance measures for Pb quantification has to be presented if applicable.
Reply: we added quality assurance measures for Pb.
3) It would be interesting if the authors provide some information on the source of the lead-contaminated kitchenware in their paper, was it purchased on the standard market or had some stranger origin.
Reply: The "indicted" pots were very old, the teflon protection very worn and the origin was uncertain (having lost the labels). According to the patient, they were a remainder of her long stay abroad Italy.
Minor comments
The paper requires a minor language correction. Also, please pay attention to the use of either British or American spelling standard, now there are some traits of both.
Reply: we hope that the English has been improved.
The sentence "This case-report is a paradigm and only the knowledge associated with a specialized laboratory and clinical experience, consequence of confidence with this kind of pathologies, allowed us to find, even if late in diagnosis, the therapy and a good resolution of the case" present in abstract and conclusion is magniloquent and unclear.
Reply: The sentence in the abstract has been lifted and in the conclusions has been partially rewritten.
L21 non exposed should be together or with a hyphen.
Reply: L21 was corrected.
L38 I guess lead is known from ancient times since it can be quite easily produced from the ore by the early history metallurgists and not because of the fact it is a bad conductor etc.
Reply: the sentence was corrected.
L45 a typo, haem
Reply: L45 was corrected.
L121 herself doesn't need a hyphen.
Reply: L121 was corrected.
L180 Avoid using contraction in an academic text (could not).
Reply: L180 was corrected.
L186 lifestyle without a hyphen.
Reply: L186 was corrected.
L190 Revise the sentence since now it is misleading and sounds as you have pots rather than data in Table 2.
Reply: 190 the sentence was corrected.

Round 2
Reviewer 1 Report
The conclusion section is the summary of this study, and it is better not to cite any references. In this section, the authors should emphasize that there are many pathways to Pb exposure. This study investigated the indoor environment, tap water, herbal medicines, cooking pots, etc., and found that Pb derived from the cooking pots is likely to be the main cause of Pb poisoning. However, other sources cannot be ruled out due to the food sources of patient are not stable. In addition, some valuble advises should be put forword for peoples in daily life.
Author Response
Reviewer 1
Open Review
(x) I would not like to sign my review report
( ) I would like to sign my review report
English language and style
( ) Extensive editing of English language and style required
( ) Moderate English changes required
(x) English language and style are fine/minor spell check required
( ) I don't feel qualified to judge about the English language and style
|
Yes |
Can be improved |
Must be improved |
Not applicable |
|
|
Does the introduction provide sufficient background and include all relevant references? |
( ) |
(x) |
( ) |
( ) |
|
Is the research design appropriate? |
( ) |
(x) |
( ) |
( ) |
|
Are the methods adequately described? |
( ) |
(x) |
( ) |
( ) |
|
Are the results clearly presented? |
( ) |
(x) |
( ) |
( ) |
|
Are the conclusions supported by the results? |
( ) |
( ) |
( ) |
(x) |
Comments and Suggestions for Authors
The conclusion section is the summary of this study, and it is better not to cite any references. In this section, the authors should emphasize that there are many pathways to Pb exposure. This study investigated the indoor environment, tap water, herbal medicines, cooking pots, etc., and found that Pb derived from the cooking pots is likely to be the main cause of Pb poisoning. However, other sources cannot be ruled out due to the food sources of patient are not stable. In addition, some valuble advises should be put forword for peoples in daily life.
Reply: Thank you very much to the reviewer for further suggestions. We are aware that reference entries in the conclusions are unsual. On the other hand, the conclusions of a case report are also a kind of discussion in which to recall the salient points of the exposure described. However we agree with the reviewer that the sources of pollution from Pb are manifold and that certainly also food (meat, probably as in the case described) or vegetables from soils polluted by industrial discharges (not only Pb, but other heavy metals such as Cd and Hg) can be a source of toxicity. Fortunately, they very rarely reach toxicity levels such as those described. The population is generally unaware of the dangers inherent in certain articles of dubious origin; it is therefore necessary that what you buy (products of any kind, edible and inedible) is certified. The text has been further expanded in this direction.
Reviewer 4 Report
Dear Editor,
I have small suggestions (see below) but these may be implemented even at the proofing stage or as a minor revision.
Sincerely,
Nikolay Solovyev
Author Response
Reviewer 4
Open Review
(x) I would not like to sign my review report
( ) I would like to sign my review report
English language and style
( ) Extensive editing of English language and style required
( ) Moderate English changes required
(x) English language and style are fine/minor spell check required
( ) I don't feel qualified to judge about the English language and style
|
Yes |
Can be improved |
Must be improved |
Not applicable |
|
|
Does the introduction provide sufficient background and include all relevant references? |
(x) |
( ) |
( ) |
( ) |
|
Is the research design appropriate? |
(x) |
( ) |
( ) |
( ) |
|
Are the methods adequately described? |
(x) |
( ) |
( ) |
( ) |
|
Are the results clearly presented? |
(x) |
( ) |
( ) |
( ) |
|
Are the conclusions supported by the results? |
(x) |
( ) |
( ) |
( ) |
Comments and Suggestions for Authors
Dear Editor,
I have small suggestions (see below) but these may be implemented even at the proofing stage or as a minor revision.
Reply: We are grateful to the reviewer for his attention.